# Salvage Therapy for Alveolar Echinococcosis—A Case Series

**DOI:** 10.3390/pathogens11030333

**Published:** 2022-03-09

**Authors:** Sanne Burkert, Lynn Peters, Johannes Bloehdorn, Beate Grüner

**Affiliations:** Department of Internal Medicine III, Division for Infectious Diseases, Ulm University Hospital, 89081 Ulm, Germany; sanne.burkert@uniklinik-ulm.de (S.B.); lynn.peters@uniklinik-ulm.de (L.P.); johannes.bloehdorn@gmail.com (J.B.)

**Keywords:** *Echinococcus multilocularis*, alveolar echinococcosis, salvage therapy, amphotericin B, benzimidazole intolerance, treatment failure, nitazoxanide, mefloquine, pembrolizumab, drug therapy

## Abstract

Benzimidazoles are the only approved drugs for the treatment of inoperable human alveolar echinococcosis but may be limited due to intolerance or, rarely, ineffectiveness. A medical second-line or salvage therapy is not available, though it is urgently needed. We report long-term follow-up data from 14 patients who underwent salvage therapy with repurposed drugs with cumulatively 53.25 patient-years. Treatment response was evaluated by both clinical outcome and image studies, preferably PET/CT. Eleven patients received amphotericin B, and 70% of evaluable cases showed some positive treatment response, but side effects often limited therapy. Five patients received nitazoxanide, of which two showed clear progression but one achieved a lasting stable disease. One patient was treated with mefloquine combination therapy in advanced disease, and overall, a positive treatment response could not be assessed. Furthermore, we report on one patient receiving pembrolizumab for a concomitant malignancy, which did not result in a reduction of echinococcal manifestation. In summary, current options of salvage therapy can sometimes induce persistent disease control, although with potentially significant side effects and high treatment costs, and mortality remains high. No clear recommendation for a salvage therapy can be given; treatment remains highly experimental, and non-pharmaceutical interventions have to be considered.

## 1. Introduction

The small fox tapeworm *Echinococcus multilocularis* (EM) is a cestode that primarily affects carnivores such as foxes and dogs as definite hosts and voles as intermediate hosts [1]. Humans can become aberrant intermediate hosts by accidentally ingesting eggs, in which case the larvae can cause alveolar echinococcosis (AE). Human AE is mainly characterised by hepatic pseudo-cystic lesions, with the potential to form metastases anywhere in the body. EM is a slowly growing organism that is rarely symptomatic at early stages and thus often diagnosed with a latency of 5–15 years. Geographically, EM is restricted to the temperate climate zone of the northern hemisphere [2]. Although AE occurs outside of tropical areas, it is considered a neglected tropical disease [3].

Untreated, the 10-year mortality can reach up to 90% [4]. The only curative therapeutic approach is complete resection of the lesions, in combination with adjuvant medical treatment. However, about 50% of patients are primarily inoperable [5], mostly due to late diagnosis associated with extensive disease manifestations and infiltration of neighbouring organs, vessels or bile ducts. In these cases, benzimidazoles (BMZs), preferably albendazole (ABZ), are the only approved medical treatment available [6]. The mode of action of BMZs is proposed to be both inhibiting parasitic tubulin polymerization as well as disrupting parasitic metabolic processes and glucose uptake [7]. Importantly, in vitro, BMZs act only parasitostatic, so a long-term, often lifelong treatment becomes necessary. Furthermore, BMZs are used peri-operatively: they are recommended for 2 years after complete resection and pre-operatively might enable resection at a later point of time or improve surgical outcome [8].

Unfortunately, side effects of BMZs are frequent, including bowel discomfort, leucopenia, alopecia and an increase in transaminases. In about 7% of patients, the increase in transaminases is dramatic, with more than 10 times of the upper normal limit, defining absolute intolerance and prompting the halt of treatment [5]. Treatment failures under BMZ therapy occur less often than side effects but can affect 1.5% to 5.2% of patients ([5], unpublished data). In these cases, no approved second-line medical treatment exists, although the clinical situation often urgently demands further options. Several substances have been reported to show anti-echinococcal properties, which can broadly be separated into anti-infectives, including antimalarial drugs, antineoplastic drugs and plant-based natural compounds [9,10,11,12]. Among these substances are both new compounds as well as repurposed drugs. The pharmaceutical development of new compounds is slow, and meanwhile, repurposing existing drugs is the only way to offer patients with BMZ intolerance or treatment failure a salvage therapy. Hitherto, only two of such substances have been used in humans and reviewed academically: amphotericin B (AMB) and nitazoxanide (NTZ).

AMB was first reported to have an in vitro parasitostatic effect on EM metacestodes in 2003, probably by interacting with membrane lipids [13]. In combination with ABZ, inhibition of the disruptive effects against metacestodes was noticed in vitro; thus, a combination therapy was not recommended [14]. A small case series of three patients from our centre who had received AMB indicated partial suppression of larval growth [15]. Another case failed to respond to a combination treatment of NTZ and AMB [16].

In addition, in 2003, NTZ proved to be effective against EM metacestodes in vitro by acting, based on different reports, either parasiticidal [17] or parasitostatic, and parasiticidal only in combination with ABZ [18]. Furthermore, NTZ showed in vivo efficacy against EM-infected mice, especially in combination with ABZ [19]. A case report from Würzburg, Germany, showed treatment failure with NTZ [16]. In another patient, peri-operative treatment did not prevent post-operative relapse [20].

Another repurposed drug investigated as salvage therapy is atovaquone, which showed in vivo activity in mice against EM, especially in combination with ABZ [21], and mefloquine, which also showed activity against murine AE but worked only when given intraperitoneally, not orally [22,23]. Furthermore, niclosamide showed good in vivo activity against EM metacestodes, but its use is currently limited by poor oral bioavailability [11]. Check-point inhibitors have also been discussed as possible immunotherapy for AE, as anti-programmed death-ligand 1 (PD-L1) treatment of infected mice leads to a reduction in parasite load [24].

In this paper, we want to give an update on AE patients treated with salvage therapy at our centre.

## 2. Results

Overall, we report 14 patients who received experimental salvage therapy since 2000, with cumulatively 53.25 treatment years at our centre (Table 1). Of note, two patients received both AMB and NTZ, explaining different case numbers in Table 1. The median age at diagnosis was 43 years (interquartile range (IQR) 23) and ranged from 7 to 75 years. Eight patients (57.2%) were male. Patients usually had extensive disease involving neighbouring organs (N1), distant metastasis (M1) or extensive hepatic disease (≥P 3), mostly resulting in stage IV AE. Regarding the reason for salvage therapy, 10 patients (71.4%) had BMZ intolerance, while 4 patients (28.6%) had BMZ treatment failure. The median time between first diagnosis and start of salvage therapy was 5 years (IQR 11), ranging from 6 months to 24 years. The median duration of salvage therapy was 2 years (IQR 3.75). Out of the 14 patients, 5 died due to AE and its complications.

In most cases, AMB was used as salvage therapy, and it is the only drug currently used for this purpose (Table 2). Eleven patients received AMB, with cumulatively 42.7 patient-years. With regard to treatment response, one patient was not evaluable, as he died before assessment due to cholangiosepsis. In seven patients (70%), we could see some positive treatment response, either by halting or by slowing down the formerly progressive disease, or even by regression. The most remarkable case was a patient with BMZ intolerance who showed stable disease (SD) for 13 years on liposomal AMB (lipAMB) every 1 to 2 weeks. When the dosage was reduced due to nephrotoxicity, the disease progressed concomitantly, so the dosage was increased again and the patient remains under close follow-up. In three patients (30%, 5.6 patient-years), we saw some kind of treatment response, but AMB needed to be stopped either due to side effects or because the patient was critically ill and it was anticipated that the drug would not be tolerated further. Currently, three patients (30%) have SD under AMB, though with a cumulative duration of treatment of only about 2 years, and the long-term outcome needs to be determined. Three patients (30%, 3.7 patient-years) showed clear progress under AMB after an average of 12 months. Notably, those patients had received lipAMB. Especially non-liposomal AMB, which was initially used, was associated with severe side effects, such as nephrotoxicity and catheter infection. With lipAMB, nephrotoxicity was still observed, but treatment was overall better tolerated, enabling a longer duration of treatment and thus better treatment evaluation. Currently, 57% of patients (n = 5) with lipAMB are showing a treatment response, while 42.9% (n = 3) are showing treatment failure. Importantly, to assess progress requires several months, as EM is a slow-growing organism, and those patients currently assessed as SD might ultimately fail treatment.

Differentiating between those having BMZ intolerance and treatment failure, no clear difference in the response to AMB was seen. With regard to serological and biochemical treatment response, in most cases, a decrease in immunoglobulin (Ig) E was correlating with SD or regression; however, the correlation was not complete. Concerning specific IgG antibodies, only a marked increase or decrease in antibodies was correlated with the clinical or radiological outcome, but the case number was too small for a proper evaluation.

NTZ was used experimentally in five patients with cumulatively 11.7 patient-years (Table 3). One patient showed clear progression, as well as hepatotoxicity; another one showed slight progress with NTZ, but SD with successfully reintroduced BMZs. A third patient had partial regression, but the medication needed to be stopped due to side effects (nephrotoxicity). In two cases, SD was achieved and structured treatment interruption (STI) was tried, but a relapse was seen in one case, after which BMZs were successfully introduced. The last case is currently stable after 11 years of STI. Overall, the experiences cannot support the parasiticidal activity of NTZ seen in vitro; however, treatment evaluation is difficult with only five patients to evaluate and an overall mixed response.

One patient received mefloquine for 8 months, initially in combination with ABZ, which was later switched to AMB (Table 3). After 7 months, FDG-PET/CT showed an increase in FDG uptake, most likely due to progressive bacterial complications and not as an indication of progressive AE; however, a positive treatment response under mefloquine could not be seen, though the ascites had diminished. Unfortunately, the patient died due to cholangitis and sepsis 1 month later in an external hospital, with all medication halted.

Furthermore, we included one patient in this analysis who received pembrolizumab (PBZ) as treatment for his relapse of urothelial carcinoma, as it is proposed as potential salvage therapy (Table 3). Prior to PBZ, he had received ABZ for 2 years for his AE, resulting in a slightly regressive disease, cystoprostatovesiculectomy and a second resection of the local relapse. Two years of PBZ did not lead to further regression, and in the latest follow-up, progression was suspected, however, only after the course of 2 years of PBZ. It is possible to suspect that PBZ suppressed the disease progression during the application; however, that is highly speculative, and most likely, PBZ did not positively affect the AE in this patient, as the patient was stable before PBZ and had even shown a reduction in FDG uptake 2 years after ABZ treatment initiation.

## 3. Discussion

The evaluation of case series of salvage therapies is difficult, as indication for alternative treatment options requires extensive, inoperable disease manifestations in which BMZs have failed either due to inefficacy or due to toxicity. Thus, situations requiring salvage therapy imply the need to treat critically ill patients with already advanced disease who are consequently more vulnerable to hepato- or nephrotoxic side effects and might ultimately die from existing AE complications, irrespective of the treatment’s effectiveness. Because of the slow nature of AE in humans, proper treatment response evaluation needs at least 6 to 12 months. Complications in between might impede evaluation, as well as treatment. Thus, the data of presented cases here have to be interpreted with caution, as the patient population is highly selected, containing the most difficult patients to treat.

AMB was the most frequently used salvage therapy. Some effectiveness could be seen, and since lipAMB is available, the treatment is better tolerated and longer follow-up was possible. However, this also revealed cases of clear progression, indicating that AMB might not be effective over time, although possibly showing an initial treatment response during the first months. Therefore, the cases we report being currently stable on AMB have to be interpreted cautiously, as the follow-up period is still short.

Even with lipAMB, nephrotoxicity and kidney failure were still observed, limiting dosage. Furthermore, treatment and opportunity costs have to be considered. The drug itself can be quite costly, and because it has to be given intravenously, mostly on a weekly basis, frequent contacts with the health system are necessary, requiring capacity and limiting the quality of life of the patient. Thus, overall, better salvage therapy options, which are not only more effective but also more tolerable, preferably oral and more readily available, are highly desirable, as many AE cases occur in countries with a less resilient health care system, which would not allow for long-term treatment with AMB.

Prior to this study, only one case of NTZ was published in detail [16], with treatment failure. Taking the five cases presented here also into account, three out of six patients showed clear progress and only one is currently showing SD with STI. Overall, from these individual reports, no conclusive judgement on effectiveness can be drawn. However, NTZ is an oral drug, and regarding side effects, it might be better tolerated than AMB and combination treatment could be useful, albeit with close monitoring of ABZ levels, since the drugs may interact. The main side effects of NTZ in our cohort were nephro- and hepatotoxicity; of the other possibly observed side effects reported, such as neurotoxicity (SAE) and reactive arthritis [20], we observed none.

Mefloquine has been argued to be a potent parasiticidal drug after in vitro results [11]. To the best of our knowledge, this is the first publication of a case treated with mefloquine, unfortunately, without a clear sign of treatment response, although it was tried in an advanced stage of disease with difficulty of evaluation. So far, the potential use of mefloquine is limited by considerable neuro-psychiatric side effects [11], which did not occur with this patient but are the reason why it was not tried, for example, as a combination therapy more often so far.

Check-point inhibition with PBZ did not lead to regression of AE lesions; however, it might be that PBZ, during application, had suppressed the progress of AE, which might have occurred due to the immunosuppression caused by the carcinoma, possibly explaining the progressive disease after stopping PBZ. However, it might also be that ending check-point inhibition itself could lead to a flare-up of AE. One way or another, it is too early to call the drug either ineffective or harmful based on one single case report and further observational data will be helpful.

Overall, none of the evaluated salvage therapies show signs of highly effective and well-tolerated second-line agents for AE. However, they are still used in desperate situations due to the current lack of other options. For the respective application, the following general lessons can be drawn from our experience with salvage therapies in human AE:(1)It seems better not to stop treatment, even if the effectiveness is questionable. In situations with advanced AE, in which treatment has been stopped completely due to complications or critically ill patients, fast progression with consecutive death was observed.(2)BMZs are currently the most effective drugs and are usually well tolerated. If not, several re-exposures with low dosages and switch from ABZ to MBZ are worth trying, as in most cases, BMZs can be introduced successfully and are then effective.(3)However, in the case of impending treatment failure or permanent intolerance, it seems best not to wait too long before starting salvage therapy, as lower effectiveness than BMZs has to be expected. A period of three, maximum six, months should be set for re-evaluation of the situation and treatment plan, with consequent escalation, if necessary.(4)Non-pharmaceutical interventions, such as a biliary stent, have to be evaluated to control complications, as well as ‘rescue’ surgery.

With respect to choosing a second-line substance, in the less frequent case of progress under BMZ therapy, our centre would currently tend towards lipAMB. In the case of absolute intolerance, we might currently try an oral option, such as NTZ, due to easier administration and probably a better profile of side effects.

## 4. Materials and Methods

All patients with salvage therapy treated at the Department of Internal Medicine III, Division of Infectious Diseases at the Ulm University Hospital, were included in this case series. The Ulm University Hospital is a tertiary care hospital and a superregional centre for AE. Diagnosis and classification were based on WHO case definition and hence included image studies ((PET/) CT, (PET/) magnetic resonance imaging (MRI) and/or abdominal ultrasound), as well as serology and, in some cases, histopathology.

Serologic tests changed over time and are reported, as used. As a screening test, an indirect haemagglutination test (IHA: Cellognost Echinococcosis, Dade Behring, Germany) or enzyme-linked immunosorbent assay (EIA: Echinococcus IgG ELISA classic ESR107G, Virion/Serion, Würzburg, Germany) was used. As a confirmatory serologic test, testing for the EM2+-antigen was used (*Echinococcus multilocularis*—ELISA Em2plus, Nr. 9300, Bordier Affinity, Crissier, Switzerland).

Treatment was according to the recommendations of the World Health Organisation Informal Working Groups on Echinococcosis (WHO-IWGE). Hepatotoxicity was defined as an increase in hepatic transaminases 5 times above the upper normal limit and absolute intolerance as a rapid increase 10 times above the upper normal limit. If repeated re-exposure with both ABZ and MBZ in reduced dosages still led to hepatotoxicity, BMZ intolerance was diagnosed. Treatment response was evaluated radiographically, preferably by PET/CT, at least every 2 years. Regression was assessed when manifestations were shrinking in size and progression if lesions were increasing. Reduction or increase in FDG uptake as such was not enough to assess progression or regression. STI was usually tried in asymptomatic patients after 2 years of BMZ treatment with negative FDG-PET and concordant serological studies. For all patients receiving salvage therapy, written consent was obtained after informing about off-label usage.

Data were analysed retrospectively from the available medical charts. Statistical analysis was conducted using Microsoft Excel 2019. In the case of non-normal distribution, the median with an interquartile range (IQR) was calculated.

## 5. Conclusions

The indication for salvage therapy in advanced AE poses a serious problem to both patients and clinicians, as no second-line agent can be recommended. Based on our data, a clear treatment recommendation cannot be deduced either. AMB has been used most frequently at our centre, yielding mixed results. NTZ has also failed in at least half of the cases, although some kind of anti-echinococcal properties could be observed in others. Due to its oral administration and less severe side effects, it could be reconsidered. However, better options are urgently needed, since currently tried salvage therapies often fail. To assess the potential benefit of the presented drugs properly, clinical studies and a patient population less sick in order to enable proper evaluation would be necessary; however, there are serious ethical restrictions for such studies.

## Figures and Tables

**Table 1 pathogens-11-00333-t001:** Baseline characteristics of reported cases.

		All Patients	AMB	NTZ
N		13	11	4
Sex	Female, n (%)	6 (46.2%)	4 (36.4%)	2 (50.0%)
	Male, n (%)	7 (53.8%)	7 (63.6%)	2 (50.0%)
Median age at first diagnosis, years (IQR)		43.0 (23.0)	43.0 (21.5)	46.0 (12.5)
Median time of salvage therapy imitation after first diagnosis, years (IQR)		5.0 (11.0)	10.0 (12.0)	8.0 (10.875)
Reason for salvage therapy	BMZ intolerance, n (%)	7 (53.8%)	5 (45.5%)	3 (75%)
	BMZ treatment failure, n (%)	6 (46.2%)	6 (54.5%)	1 (25%)
Cumulative time of salvage therapy, years		50.8	42.7	9.2
Median duration of salvage therapy		2.0 (3.75)	1.5 (3.0)	1.7 (1.0)

**Table 2 pathogens-11-00333-t002:** Patients with AMB.

Diagnosis	WHO Stage, Case Definition	Manifestations	Age at Diagnosis, Gender	Surgical Treatment	Medical Treatment before AMB	Time of AMB after Diagnosis	Duration of AMB	Reason for AMB	Initial Application	Side Effects of AMB	Serological Outcome	Radiological and Clinical Outcome
1976	P?N1M1, stage IV, confirmed	Liver, lung, CNS, mamma, paracardial	42, F	Partial liver resection 1976, CNS resection 1976	ABZ, MBZ (few months), Interferon γ (1.5 years), praziquantel (7 months)	24 years	3 years	BMZ intolerance	Non-liposomal AMB 0.5 mg/kg BW/3×/week	Nephrotoxicity, ECC reduction	Stable IHA-AB, partial reduction IgE	Stable with AMB, but advanced disease, side effects require dose reduction, deceased at 69 due to AE-related heart failure
1992	PxN1M0, stage IIIB, confirmed	Liver continuing to thoracic wall (post-operative relapse), subcutaneous manifestation, paracardial	49, M	Left hemihepatectomy 1992, subxiphoid resection 1998	ABZ (1 year), MBZ (1 year), NTZ (9 months)	15 years	16 years	BMZ intolerance	i.v. AMB-D 0.8 mg/kg BW 3×/week, changed to lipAMB 3 mg/kg every 1–3/weeks	Nephrotoxicity requiring dose reduction	Increase in IHA AB and slightly IgE concomitant with radiological progress	Long, stable disease, progress with dose reduction after 13 years
1992	P?N1M1, stage IV, confirmed	Liver, lumbar spine, retroperitoneum, CNS (intrathecal lesions)	43, M	Right hemihepatectomy 1992, vertebrectomy 1995, laminectomy and myelon decompression 2000	ABZ (12 years), MBZ (3 years)	12 years	13 months	BMZ treatment failure	Intrathecal AMB-D 0.2 mg 3×/week	Catheter infection	Not evaluable	Partial regression, stop due to catheter explantation, deceased at 57 due to sepsis with ABZ
1992	P?N?M1, stage IV, confirmed	Extensive liver, hyper-IgE syndrome	7, M	Partial liver resection 1993 and 1999	MBZ (10 years), praziquantel (pre-operatively 1999)	10 years	1.5 years	BMZ treatment failure	i.v. AMB-D 0.5 mg/kg BW 3×/week, reduced to 2×/week	Nephrotoxicity requiring dose reduction and pausing	Not evaluable	Less progressive disease under AMB, repeated stops with critical illness (cholangitis), deceased with liver failure/cholangitis
1997	P4N1M0, stage IV, probable	Liver with extended portal vein and inferior vena cava thrombosis, hepatic encephalopathy	28, F	Partial liver resection 1997 (R2), listed for transplant, ultima ratio explorative laparotomy and partial cystectomy February 2021	ABZ (11 years), MBZ (4 months)	23 years	17 months in combination with ABZ	BMZ treatment failure	lipAMB 3 mg/kg BW/week *	Nephrotoxicity	Slight increase in IgE, stable IgG AB, borderline IHA	Progress under AMB, post-operative death
2009	P4N1M0, stage IV, confirmed	Disseminated liver with secondary sclerotic cholangitis	57, M	Inoperable, listed for transplant	ABZ (10 years)	10 years	5 months in combination with mefloquine	BMZ treatment failure	lipAMB 3 mg/kg BW/week *	Nephrotoxicity	EM-AB IgG stable	Deceased due to cholangitis, AMB not evaluable
15 Dec	P4N1M0, stage IV, confirmed	Right liver lobe, probably kidney	75, M	Inoperable	ABZ (6 months), MBZ (1 months)	5 years	10 months	BMZ intolerance	lipAMB 3 mg/kg BW/week *	nephrotoxicity	Decrease in IgE, EM-AB IgG constantly high	Stable disease
18 Nov	P3N0M0, stage IIIa, confirmed	Right liver lobe with contact to right portal vein	39, M	Extended right hemihepatectomy November 2020	ABZ (3 months), MBZ (1 months)	1 year	15 months	BMZ intolerance	lipAMB 3 mg/kg BW/week *	-	Decrease in EM-AB IgG, post-operatively negative	Pre-operative progression with AMB, currently post-operative remission with AMB
19 May	P4N1M0, stage IV, confirmed	Liver with portal vein thrombosis, stenosis of DHC, left renal vein, truncus coeliacus and AMS	60, F	Palliative surgery with debulking 09/21	ABZ (4 months), MBZ (4 months)	2 years	8 months	BMZ intolerance	lipAMB 3 mg/kg BW/week *	-	Post-operative decrease in IgE, constant EM-AB IgG	Stable disease
19 Nov	P4N1M1, stage IV, probable	Disseminated liver, spleen, peritoneum	59, M	Inoperable	ABZ (5 months), MBZ (2 months)	1 year	1 year	BMZ intolerance	lipAMB 3 mg/kg BW/week *	-	Strong increase in EM-AB IgG, slight decrease IgE	Progress, currently re-exposure MBZ
21 Jan	P4NxMo, stage IIIB, confirmed	Left liver lobe with life-threatening cholestasis and cholangitis	34, F	Inoperable	ABZ (10 months)	8 months	5 months in combination with ABZ	Combination therapy due to life-threatening condition, BMZ treatment failure and intolerance	lipAMB 3 mg/kg BW 3 times/week *	-	Decrease in EM-AB IgG and normalisation of IgE	Stable disease with combination treatment, re-evaluation operation planned

* After loading with 3 mg/kg body weight lipAMB for 10–14 days.

**Table 3 pathogens-11-00333-t003:** Patients with nitazoxanide, mefloquine and pembrolizumab.

Nitazoxanide												
Diagnosis	WHO Stage, Case Definition	Manifestations	Age at Diagnosis, Gender	Surgical Treatment	Medical Treatment before NTZ	Time of NTZ after Diagnosis	Duration of NTZ	Reason for NTZ	Dosage	Side Effects of NTZ	Serological Outcome	Clinical Outcome
1992	PxN1M0, stage IIIb, confirmed	Liver continuing to thoracic wall (post-operative relapse), subcutaneous manifestations, paracardial	49, M	Left hemihepatectomy 1992, subxiphoid resection 1998	ABZ (1 year), MBZ (1 year)	13 years	9 months	BMZ intolerance	2 × 500 mg/day (4 months), reduced to 2 × 500 mg/week	Hepatotoxicity	Increase in IHA AK, IgE within normal limits	Progression, severe hepatotoxicity limited dosage
1992	P?N1M1, stage IV, confirmed	Liver, lumbar spine, retroperitoneum, CNS (intrathecal lesions)	43, M	Vertebrectomy 1995, laminectomy and myelon decompression 2000	ABZ (12 years), MBZ (3 years), AMB-D (13 months)	14 years	17 months	BMZ treatment failure	2 × 500 mg/day	Nephrotoxicity, interaction with ABZ (increased serum levels of ABZ)	Partial regression of IHA AK	Partial regression, stopped due to acute chronic kidney failure, deceased at 57 due to sepsis, with ABZ
3 June	P3N0M0, stage IIIa, confirmed	Right liver lobe	17, F	Right hemihepatectomy 01/04	ABZ and MBZ (together 4 months)	6 months	2 years	BMZ intolerance	Cyclical 2 × 1 G/day for 30 days and 3 months break peri-operatively for 6 cycles (2 years)	-	Increase in IHA AK concomitant with relapse	Relapse 1 year after stopping NTZ with BMZ (9 years) regression, currently on STI
3 Dec	P3N0M0, stage IIIa, probable	Extended liver disease	49, F	Inoperable	ABZ and MBZ (together 2 years on/off)	3 years	5 years (initial attempt to combine with BMZ failed)	BMZ intolerance	500 mg BID	-	Decrease in IgE, constantly high IHA AK	Partial regression and stable disease since STI 11 years ago
4 June	P3N1M0, stage IIIb, probable	Extended liver disease	22, M	Inoperable	ABZ (2 years)	2 years	2.5 years (1.5 in combination with ABZ)	BMZ intolerance	500 mg BID	Gastrointestinal side effects	Increase in IHA with monotherapy, decrease in IgE with either treatment	Slight progress with NTZ monotherapy, stable with combination therapy and ABZ monotherapy
Mefloquine												
2009	P4N1M0, stage IV, confirmed	Disseminated liver with secondary sclerotic cholangitis	57, M	Inoperable, listed for transplant	ABZ (10 years)	10 years	8 months (in combination with ABZ or AMB)	BMZ treatment failure	250 mg/week	Nephrotoxicity	EM-AB IgG stable	Deceased due to cholangitis, no treatment response seen after 7 months
Pembrolizumab												
16 Feb	P4N1?M0, stage IV, confirmed	Right liver lobe	64, M	Diagnostical laparotomy and partial resection 02/16	ABZ	2.5 years	2 years	Urothelial carcinoma	200 mg every 3 weeks, followed by 400 mg every 6 weeks	Dermatotoxicity	Increase in IgE	Suspicion of progressive AE

## Data Availability

Not applicable.

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
