# Peer review of "Salvage Therapy for Alveolar Echinococcosis—A Case Series"

_pathogens, 2022, doi:10.3390/pathogens11030333_

Round 1

Reviewer 1 Report

From my point of view the topic presented in Your manuscript is valuable and interresting described. In the literature there is small amount  numbers of cases  of  advanced alveococcosis. Treatment is difficult and unfortunately we do not have appropriate tools to achieve suitable results. All situations, in which you can introduce new methods are worthy highlited.

I think Your work can be published in  the Pathogens

Author Response

We want to sincerely thank reviewer 1 for his positive vote for our paper.

Reviewer 2 Report

The manuscript entitled: “Salvage Therapy for Alveolar Echinococcosis” addresses a highly interesting topic. Until now, benzimidazoles are considered the main drug against Echinococcosis. However, this chemical is not advisable in the long term owing. The study is interesting and will be benefit for exploring novel drugs against Echinococcosis. However, I am afraid that I will not refer to such report. The article has in my opinion a low value and does not represent a contribution to the field. And I think this is a case series article.

All in all, I suggest that the authors could greatly enhance the usefulness of this work for the community if they consider the following changes:

  • Key words: should be revised. Some words are repeated in the title. Also please use the Mesh database for keywords.
  • It is best to provide a list of abbreviation.
  • Some words, especially the names of drugs, are misspelled (for example line 12: Nitaxozanide). Please recheck all the text.
  • In general, the method material needs to be rewritten and completed. Also, the inclusion and exclusion criteria should be mentioned.
  • The title of Table 1 has been moved and is not seen.
  • Please recheck the number of patients in the tables. For example, the sum of patients in Table 1 is not 13 (11+4)???
  • Ethical approval of the study is not mentioned.
  • Conclusions can be improved.

Round 2

Reviewer 2 Report

Accept